# Cytogeography of Naturalized *Solidago canadensis* Populations in Europe

**DOI:** 10.3390/plants12051113

**Published:** 2023-03-01

**Authors:** Zhongsai Tian, Jiliang Cheng, Jingxuan Xu, Dongyan Feng, Jian Zhong, Xiaoxiao Yuan, Zheng Zhang, Yu Zhang, Zhiyuan Mao, Sheng Qiang

**Affiliations:** Weed Research Laboratory, College of Life Sciences, Nanjing Agricultural University, Nanjing 210095, China

**Keywords:** chromosome number, DNA content, geographical differentiation, invasive plant, molecular marker, *Solidago altissima*

## Abstract

Autopolyploidization has driven the successful invasion of *Solidago canadensis* in East Asia. However, it was believed that only diploid *S. canadensis* invaded Europe, whereas polyploids never did. Here, molecular identification, ploidy level, and morphological traits of ten *S. canadensis* populations collected in Europe were compared with previously identified *S. canadensis* populations from other continents and *S. altissima* populations. Furthermore, the ploidy-driven geographical differentiation pattern of *S. canadensis* in different continents was investigated. All ten European populations were identified as *S. canadensis* with five diploid and five hexaploid populations. Significant differences in morphological traits existed among diploids and polyploids (tetraploids and hexaploids), rather than between polyploids from different introduced ranges and between *S. altissima* and polyploidy *S. canadensis*. The invasive hexaploids and diploids had few differences in latitudinal distributions in Europe, which was similar to the native range but different from a distinct climate-niche differentiation in Asia. This may be attributed to the bigger difference in climate between Asia and Europe and North America. The morphological and molecular evidences proved the invasion of polyploid *S. canadensis* in Europe and suggest that *S. altissima* may be merged into a complex of *S. canadensis* species. Our study may be concluded that geographical and ecological niche differentiation of an invasive plant driven by ploidy depends on the degree of difference in the environmental factors between the introduced and native range, which provides new insight into the invasive mechanism.

## 1. Introduction

*Solidago canadensis* L. is an invasive plant native to North America and is now widely distributed in Asia, Europe, and Australia [1,2]. It is a polyploid complex (*S. canadensis* L. complex) composed of multiple ploidy subspecies or variants, with three main ploidy levels in its native and introduced range: diploid (2n = 18), tetraploid (2n = 36), and hexaploid (2n = 54) [3,4,5,6], all of which are harmful weeds [6,7,8]. The hybridization, variable chromosome number, and morphological characters of genus *Solidago* make its classification extremely complex, and distinguishing them can be based on only a few quantitative traits [3,9]. Among them, the taxonomic status of *S. altissima* and *S. canadensis* has been controversial, with some arguing that the two should be one species, and *S. altissima* should be considered as a variety of *S. canadensis* [10,11,12], and some suggesting that *S. altissima* is a separate species by morphological analysis [9,13]. Most European populations of both species are described as *S. altissima* [1,12], but macro-morphological analysis suggests that *S. canadensis* is a common species in Europe [13,14]. The main differences between the two are the length and arrangement of epidermal hairs on stems and leaves, the shape and margins of leaves, and micromorphological features of the leaf epidermis and the rhizome system [15], but such distinguishing traits are sometimes inaccurate in complex environments. In Europe, *S. canadensis* can hybridize with native *S. virgaurea* L., giving the hybrid *S. ×niederederi* Khek [16], which can be identified by molecular tools [17]. The development and application of molecular markers can help elucidate intraspecific or interspecific evolutionary relationships, taxonomy, and rates of genomic change [18], but there are no systematic molecular markers for *S. canadensis* and its complex species [6,19].

Polyploidy is a key factor in the rapid adaptive evolution of invasive plants [20,21,22,23,24]. Polyploids are likely to compete with diploids when they emerge, but they often drive ecological niche shifts to prevent this competition [25,26]. An increasing number of research suggests that autopolyploidization facilitates plants to expand their distribution and invade new environments by generating new traits [20,22]. Polyploid plants typically have higher biomass, larger rhizome systems, larger flowers, and more seeds [22]. For example, the introduced tetraploid of *Solidago gigantea* Aiton was taller and produces more rhizomes than the native diploids [8]. A study on *Centaurea stoebe* L. showed that the fruiting rate of tetraploid plants was significantly higher than that of diploids, with the highest fruiting rate in the introduced tetraploids [27]. The high genetic variability of polyploid plants improves their ability to adapt to new environments, making them more likely to become invasive plants [28,29]. Some autopolyploids have greater environmental tolerance, such as cold tolerance [30], heat tolerance [31], salt tolerance [32], and drought tolerance [30]. Studies have shown that the invasive ability of plants depends on the ploidy level and tolerance to environmental factors, with polyploids and species with enhanced tolerance to high temperature and rainfall having a higher invasive ability [33].

Several invasive plants will grow in areas with a similar climate to their native range after invasion, i.e., ecological niche conservatism [34]. However, many invasive plants change their climatic niche after the invasion [34,35], and multiple factors, including human activities and climate change, may facilitate plant adaptation to new climatic conditions, leading to a shift in the climatic niche between the native and introduced range [36]. The increase in genome size caused by polyploidization has important significance for the ecological niche spread of invasive plants [37,38]. For example, *Centaurea maculosa* Lam. has ploidy-driven geographical divergence in its native Europe, but mainly tetraploid populations have been introduced into North America and its ecological niche has changed compared to the native tetraploids [26,39,40]. There is also a clear geographical differentiation of *S. gigantea* in its native range, but only tetraploid populations are found in the introduced range of Europe and Asia [8,41,42]. These studies generally compared ecological niche differentiation only between the native and part of the introduced range, and because there were no diploid populations in the introduced range, it was not possible to compare whether ploidy-driven patterns of geographical differentiation remained consistent between the native and introduced range.

Cheng et al. [6] identified 152 populations of *S. canadensis* of different geographical origins worldwide by ribosomal ITS and *trn*H-*psb*A intergenic spacer, and determined the ploidy level of each population. Results showed that diploid, tetraploid, and hexaploid populations were present in both North America and Asia, but only diploids in Europe. Polyploidy has contributed to the invasion and geographical differentiation of *S. canadensis* in East Asia [6]. Polyploid populations have stronger antioxidant systems that help them scavenge reactive oxygen species at high temperatures, resulting in greater heat tolerance and facilitating their expansion into subtropical regions [6]. There are differences in morphological traits among different ploidy populations, and polyploids usually are taller, have more biomass and rhizomes, and can produce larger amounts of seeds [6,43,44]. Polyploids have stronger adaptive and competitive capacities than diploids [43,45], and are likely to expand into wider adaptive environments and regions under global climate change. However, the existence of polyploid *S. canadensis* in Europe is unclear. *S. canadensis* and *S. altissima* expanded rapidly after their introduction to Europe in the 17th–18th century, mainly distributed in Central and Eastern Europe, with a trend of southward expansion [1,46,47,48]. It can be found along roadsides in many countries in Europe [49]. Previously, only diploid populations were found in both species in Europe [5,6,12,16,47,50,51]. Verloove et al. found hexaploid *S. altissima* in Belgium and suggested that all previously reported diploid populations were *S. canadensis* [13,52]. However, given the uncertainty of the distinction between *S. altissima* and *S. canadensis*, this population may be also a polyploid population of *S. canadensis*.

Hence, molecular marker and ploidy determination were performed on ten new populations in Europe to determine whether these populations are *S. canadensis*, as well as their ploidy level. Molecular identification results and morphological traits of *S. canadensis* populations and populations previously identified as *S. altissima* by phenotypic characteristics were compared to clarify the relationship between them. The morphological differences of different geo-cytotype populations were compared. In addition, we investigated the niche differentiation of different ploidy populations in Europe, and compared the pattern with those in North America and Asia, to further improve its geographical distribution pattern in the Northern Hemisphere and invasion mechanism.

## 2. Results

### 2.1. Molecular Markers and Phylogenetic Analysis

The results of molecular markers showed that all ten populations from Europe were *S. canadensis*, and the ribosomal ITS sequences of EU10, EU11, and EU14 had one base mutation each compared to the sequence from GenBank, while the sequences of the other seven populations were completely consistent with *S. canadensis* (Table 1); the *trn*H-*psb*A intergenic spacer of EU02 had one base mutation, while the sequences of the other nine populations were completely consistent with *S. canadensis* in GenBank (Table 2).

The ribosomal ITS and *trn*H-*psb*A intergenic spacer of the populations phenotypically identified as *S. altissima* were > 99% similar to *S. canadensis* (Appendix A). Therefore, we preliminarily consider that *S. canadensis* and *S. altissima* are one species by molecular level. The ribosomal ITS and *trn*H-*psb*A intergenic spacer of 159 populations were spliced to construct a phylogenetic tree, and all populations except RUS02 were clustered together with the sequences of *S. canadensis* from GenBank and separated from other closely related species (Figure 1). The *S. canadensis* populations from Europe, other *S. canadensis* populations, and *S. altissima* populations were clustered together in the phylogenetic tree (Figure 1).

### 2.2. Ploidy Level of S. canadensis Populations in Europe

The peak positions of five populations in Europe were 2.6 to 2.7 times that of the internal reference diploid population, all of which should be considered as hexaploidy [6,13,53]; five overlapped with the peak of the diploid population, which were diploid (Figure 2b–k; Appendix A). Based on the results of molecular markers and ploidy determination, we concluded that these populations in Europe consisted of five diploid populations and five hexaploid populations of *S. canadensis*.

### 2.3. Comparison of the Morphological Traits of Different Geo-Cytotype Populations

The plant height, stem diameter, leaf length, and leaf width of the introduced populations were significantly higher than those of the same ploidy of native populations (*p* < 0.05), and the polyploids were significantly higher than the diploids (*p* < 0.05; Table 3; Appendix A). The number of epidermal hairs of native populations was significantly higher than that of the introduced populations of the same ploidy (*p* < 0.05), but there was no significant difference between different ploidy in both the native and introduced range. The number of ray florets and disc florets did not differ significantly between different geo-cytotypes, but the length of the ray floret and height of involucre were significantly higher in the polyploids than in the diploids (*p* < 0.05; Table 3). A comparison of the aspect ratio of leaves and the ratio of plant height to stem diameter at the adult stage showed that the leaves were narrower in the native populations than those of the same ploidy of introduced populations, and narrower in the diploids than polyploids (*p* < 0.05; Figure 3a,b). The ratio of plant height to stem diameter was greater in the native populations than those of the same ploidy in the introduced range (*p* < 0.05), and the diploids were significantly greater than the polyploids (*p* < 0.05; Figure 3a,b and Appendix A). This suggests that the differences in morphological characters of different *S. canadensis* populations are mainly caused by both cytotype and geographical factors.

The polyploid populations from different introduced regions were not significantly different for each trait, but were significantly different from the native polyploids (*p* < 0.05; Table 3). Based on the cluster analysis of 11 phenotypic traits, the North American polyploid population, the Asian polyploid population, the European hexaploid population, and *S. altissima* were clustered together as one group, where the introduced and native polyploid populations divided into two branches. The Chinese cultivated species “huangyinghua”, the European diploid populations, the Russian diploid population, and the native diploid populations clustered together. The *S. canadensis* populations were separated with the closely related species *S. gigantea*, *S. simplex,* and *S. decurrens* (Figure 3c). It showed that there were no significant differences in a biological trait between the polyploids of *S. canadensis* and *S. altissima*, and the cluster analysis also indicated that the polyploids of *S. canadensis* and *S. altissima* clustered together (Table 3; Figure 3). These results further suggest that *S. canadensis* and *S. altissima* belong to the same species.

### 2.4. Geographical Distribution in Europe and Global Differentiation of Ecological Niches of Different Ploidy Populations

We have previously demonstrated that *S. altissima* and *S. canadensis* are one species by molecular markers and comparison of morphological traits (Figure 1; Appendix A), and therefore integrated *S. altissima* in the literature into *S. canadensis* species for analysis. The results showed that the diploid population in Europe was mainly distributed in central Europe, while the hexaploid was relatively widely distributed, with a relatively wide range of latitudes and temperatures (Figure 4A,B). The latitudes of the two ploidy populations did not differ in general, but the mean July temperatures of the distribution areas differed, ranging from 15 to 21 °C for diploids and 17 to 25 °C for hexaploids (Figure 4B,C).

No significant geographical differentiation was observed between diploid and polyploid populations in Europe, as in North America, while a highly significant geographical differentiation was found between the diploid and polyploid populations in Asia, with polyploids distributed at lower latitudes (Appendix A and Figure 5a). However, the mean July temperature was significantly higher in the polyploid distribution than in the diploid, both in the native and different introduced ranges (Figure 5b).

## 3. Discussion

### 3.1. The Relationship of S. canadensis and S. altissima

Verloove et al. [13] found hexaploid *S. altissima* in Europe and identified it as different from *S. canadensis* mainly based on morphological comparison and ploidy determination. The study used diploid *S. canadensis* as a reference, whereas our study showed that morphological differentiation between diploids and polyploids of *S. canadensis* was obvious, and together with the absence of molecular marker identification, the results of this study are difficult to confirm. Semple and Uesugi [55] compared *S. altissima*, *S. canadensis*, *S. chilensis,* and *S. gigantea* for phenotypic comparison, where the phenotypic characters of *S. altissima* and *S. canadensis* were partially overlapping, but the study also did not distinguish ploidy levels. It has been shown that *S. altissima* and *S. canadensis* do not differ in functional traits, biomass production and allocation, and ploidy levels [47,56,57]. *S. canadensis* and *S. altissima* were also not significantly different in their ecological distribution in Europe [15]. The main difference between *S. canadensis* and *S. altissima* is the length and arrangement of the epidermal hairs of the stems and leaves [15]. However, the present study (Table 3; Figure 3) and several previous studies have shown that the differences in phenotypic traits among different populations were influenced by ploidy and environment [43,44]. These findings, in combination with the results of this study, suggest that these phenotypic differences between *S. altissima* and *S. canadensis* result from polyploidy and the evolution of adaptation to the environment.

Furthermore, the ribosomal ITS and *trn*H-*psb*A intergenic spacer sequence similarity between *S. canadensis* and *S. altissima* was greater than 99%, and the DNA content of hexaploid *S. canadensis* was the same as that of hexaploid *S. altissima* found in Europe (Figure 1) [13]. The DNA content of the diploid *S. canadensis* found by Szymura et al. [47] was also not significantly different from that of the diploid *S. altissima*. Morphological comparisons showed that the differences in morphological characters between polyploid *S. canadensis* and *S. altissima* were not significant (Table 3; Figure 3). In summary, we concluded that *S. canadensis* and *S. altissima* should be considered as one complex species (*S. canadensis* L. complex) from the comparison of molecular identification, morphological characters, and DNA content.

### 3.2. Ploidy and Environment Determine the Morphological Diversity of S. canadensis

The growth and morphological traits of plants are important indicators for the efficient utilization of resources and tools to indicate the competitive ability of plants [58]. The general effect of polyploidy is an increase in plant size; polyploids typically have taller, more robust plants that produce higher biomass, larger rhizome systems, and larger flowers and seeds [59,60,61,62]. For example, introduced tetraploids of *S. gigantea* were taller than the native diploids [8]. Introduced polyploids of *Fallopia* have been successfully invaded due to taller plants, higher biomass, and larger rhizomes [59]. Our results showed that plant height, stem diameter, leaf length, and leaf width were significantly higher in polyploids of *S. canadensis* than in diploids in different regions (Table 3). Epidermal hairs can resist biotic and abiotic stresses in the environment [63]. *S. canadensis* has a variety of natural enemies in its native range, but lacks natural enemies in the introduced range [11,64], so the native *S. canadensis* may resist insects and other enemies by having more epidermal hairs. The polyploid *S. canadensis* had a larger flower head than the diploids due to polyploidization (Table 3). The flowers of tetraploids and triploids of *Leptospermum scoparium* JR and G. Forst were significantly larger in diameter than diploids [65], and the flowers of *Escallonia rubra* tetraploids were also larger than diploids [66]. The greater flower heads help to attract pollinators, increase the probability of cross-pollination, ensure sufficient seed production, and improve the adaptability of *S. canadensis* [67]. Differences in phenotypes of *S. canadensis* between geo-cytotypes were observed, with the native polyploid being taller and having a more robust stem than the diploids, while the introduced polyploids had taller and stronger stems than the native polyploids (Figure 3), suggesting that *S. canadensis* has enhanced growth capacity through pre- and post-adaptation, resulting in greater invasive ability.

Polyploids can promote biological invasion by accelerating the differentiation of morphological traits through pre- and post-adaptive evolution [22]. Invasive plants usually have characteristics such as high phenotypic plasticity, high reproductive capacity, and suitability for dispersal [68]. The change in phenotypic expression of a genotype in response to environmental factors is known as “phenotypic plasticity” [69,70]. In Europe, *S. canadensis* has a relatively wide niche with regard to soil properties, which can be related to both local adaptation and phenotypic plasticity [56,71]. Studies have proved that plasticity in *S. canadensis* plays a central role in shade adaptation [72] and salt tolerance [73], but these studies did not compare phenotypes of different ploidy populations. Our study and Cheng et al. [6] proved that the polyploids had a higher phenotypic plasticity than diploids. Studies on *S. gigantea* have shown that only tetraploids have successfully invaded Europe, that native tetraploids produce more branches and rhizomes than diploids, and that population size, density, and total plant biomass were greater in introduced populations than in native populations [8,74]. A study of *Mimulus guttatus* DC. showed that there were no significant differences in phenology, floral traits, sexual, and nutritional reproduction between the North American (native range), Scottish, and New Zealand populations (introduced range), but the introduced populations had twice as many flower-bearing upright side branches as those of native populations, due to the fact that different introduced populations have evolved by adaptation and enhanced reproduction [75]. The morphological characteristics of polyploid populations of *S. canadensis* in different introduced ranges did not differ significantly from each other, but they differed from the native populations, especially diploid populations (Table 3), indicating that polyploids underwent rapid adaptive evolution after the invasion in different regions, adapted to the local ecological environment, and finally invaded successfully.

### 3.3. Polyploid S. canadensis Has Been Successfully Invaded and Widely Distributed in Europe

Studies have investigated the cytogeography patterns of *S. canadensis* in its native range in North America, and Asia where it is invasive, and diploids, tetraploids and hexaploids were found [6,76]. However, few studies have investigated the distribution of ploidy levels in *S. canadensis* in Europe, and previous studies have suggested that only diploid is present in Europe [5,6,16,47,50]. In general, diploids are less abundant in the invasive range than polyploids and are present in areas where climatic conditions are similar to those of native populations [77]. For example, there are diploids, tetraploids, and hexaploids of *S. gigantea* in the native range, but only tetraploids have invaded Europe [41]. Studies on *C. maculosa* have also shown that the proportion of polyploids is much higher in North America, where it is more invasive than in its native range in Europe [40]. Previous studies have shown that only polyploids of *S. canadensis* have successfully invaded in Asia [6], and five hexaploid populations and five diploid populations were found in Europe in our study (Figure 1), suggesting that polyploids may also have been widely distributed in Europe.

The polyploid populations of *S. canadensis* in Asia have changed their isothermal niche by adapting to summer heat stress and are distributed at a lower latitude compared to diploids [6]. Distributions of *S. canadensis* were limited climatically in Central Europe, and were also correlated with proxies of human pressure [78]. The latitudinal difference between diploid and hexaploid populations found in Europe was not significant, but hexaploids have a wider latitudinal range of distribution (Figure 3). Heat tolerance is usually a key factor for the successful invasion of invasive plants, and greater heat tolerance of polyploids could help them expand their invasion range [33]. The mean July temperature in the distribution areas of hexaploids was significantly higher than that of diploids (Figure 3d), suggesting that hexaploids in Europe have also evolved adaptively and are better adapted to temperature changes. Currently, *S. canadensis* is widely distributed in central Europe [47,52,57], and some studies also predict that *S. canadensis* will invade further south in Europe [46], which may be related to the greater adaptability and heat tolerance of polyploids [6]. Certainly, the number of populations we studied is not large enough, so future attention should be paid more to the ploidy distribution of *S. canadensis* in Europe and its invasion in the south.

### 3.4. Ploidy-Driven Ecological Niche Differentiation of S. canadensis in the Northern Hemisphere Depends on Environmental Factors

*Solidago canadensis* is widely distributed in North America, including the temperate continental climate zone in the northeast and the subtropical humid climate zone in the southeast [6]. The geographical distribution and ecological niches of diploids and polyploids have diverged significantly after the invasion, which is mainly due to the deviation of the climatic ecological niches of polyploids. Diploids are still distributed in regions with similar climates to their native regions, such as Europe and the Russian Far East. Polyploids have invaded China and Europe, mainly distributed in the subtropical monsoon climate zone in China, and in the Europe range with a climate similar to that of North America (Figure 5). This is mainly due to the weak adaptation of diploids, while polyploids are more environmentally adapted and can successfully invade a wide range of climatic regions [79].

Whether invasive plants show ecological niche differentiation after the invasion is influenced by several factors [80,81]. Polyploidy can facilitate the adaptive evolution of plants, which is important for the spread of plant invasions [23,82]. There is no generalized pattern of niche differentiation during the establishment and evolution of polyploids, and different lineages of the same polyploid plant may coexist with different patterns of climatic niche evolution (niche conservatism, contraction or expansion) [37]. Whether polyploids develop niche differentiation depends on the history of the species: e.g., age of the polyploid, multiple origins, or the number of ploidy levels, among others [83]. In addition, the climate is the main factor driving the distribution of plant species [84], which has an important impact on the ecological adaptability of different cytotypes [85]. *S. canadensis* invaded East Asia and Europe at about the same time, but the geographical differentiation patterns of different ploidy populations after the invasion were different. The climate of the distribution area of *S. canadensis* polyploids in China was different from that in North America and Europe, and the mean July temperature was significantly higher than that of the polyploid population in North America and Europe (Figure 5). Polyploids in East Asia have evolved through adaptation to enhance heat tolerance to expand into subtropical regions, while diploids are unable to adapt to high temperatures and can only be distributed in mid-latitudes, thereby promoting geographical differentiation of ploidy [6]. As the climate was similar to that of the native range, there was also no clear ploidy-driven geographical differentiation in Europe. In summary, we suggest that it is mainly climatic factors that lead to the adaptive evolution of polyploids of *S. canadensis* in different introduced ranges, hence to different patterns of ecological niche differentiation. Our study indicated that the ploidy-driven adaptive evolution of invasive plants depends on the degree of difference between the environmental factors of the introduced range and the native range.

Even though there was no difference in latitude between the distribution of different ploidy populations in North America and Europe, the mean July temperature of polyploids was still higher than that of diploids (Figure 5), indicating that there is a difference in the required optimum ecological niches of different ploidy populations, and temperature is an important factor influencing the geographical distribution of *S. canadensis*. Under the environmental conditions of global warming, the future distribution of *S. canadensis* is likely to further expand from the temperate zone to pantropical regions driven by polyploidization.

## 4. Materials and Methods

### 4.1. Experimental Materials

The ten populations collected from Europe (Appendix A) were performed for ploidy determination and molecular identification. Molecular identification results were used to compare and construct phylogenetic trees together with 142 *S. canadensis* populations from other regions and seven populations identified as *S. altissima* by phenotypic characteristics as previously described by Cheng et al. [6].

Biological characteristics were measured for ten populations from Europe and 93 other different ploidy populations of *S. canadensis* collected from the native regions (US and Canada) and introduced regions (China, Russia), and 8 populations identified as *S. altissima*, and one population each of *S. gigantea*, *S. simplex*, and *S. decurrens* (Appendix A). The detailed descriptions of material collection and identification had been presented by Cheng et al. [6].

The ‘NA2x’, ‘NA4x’, ‘NA6x’, ‘IN2x’, ‘IN4x’, and ‘IN6x’ were used to represent the native diploid, native tetraploid, native hexaploid, introduced diploid, introduced tetraploid, and introduced hexaploidy of *S. canadensis*, respectively.

### 4.2. DNA Extraction and Molecular Identification

To confirm the correct identification of the ten populations, we sequenced the ribosomal ITS (608 bp) and *trn*H-*psb*A intergenic spacer (213 bp) of three individuals from each population. To this end, the total DNA from the dried leaves was extracted using a Plant Genomic DNA kit (DP305, Tiangen Biotech Co., Ltd., Beijing, China). A pair of primers (upstream: 5′-CGTAACAAGGTTTCCGTAG-3′, downstream: 5′-TTATTGATATGCTT AAACTCAGCGGG-3′) were used to amplify ITS rDNA gene and another pair of primers (upstream: 5′-CCGCCCCTCTACTATTATCTA-3′, downstream: 5′-TCTAGACTTAGCAGCTATTG-3′) were used to amplify *psb*A-*trn*H intergenic spacer. Polymerase chain reaction (PCR) was carried out in 50 μL containing 2 μL of 10 μmol/L primers, 10 × buffer 5 μL, 25 mM Mg^2+^ 4 μL, 2.5 mmol/L dNTPs 4 μL, 50 ng/μL DNA templates 1 μL, and 5U Taq DNA polymerase 0.5 μL. The PCR cycle was 95 °C for 4 min followed by 30 cycles of 95 °C for 30 s, 52 °C for 60 s, 72 °C for 60 s, and a final extension step at 72 °C for 7 min. PCR products were separated on 1% agarose gels and stained with ethidium bromide. The PCR products were purified and sequenced.

### 4.3. Construct Phylogenetic Tree

The ribosomal ITS and chloroplast *psb*A-*trn*H were sequenced and edited using the BioEdit v 7.0.5.3 software. As the reference for alignment, we extracted the data for *S. canadensis* (ITS, HQ142590.1; *psb*A-*trn*H, KX214929.1) from GenBank and aligned them with the ITS and *psb*A-*trn*H sequences of our samples using the Clustal X 1.83 software. The ITS1, 5.8S, and ITS2 gene boundaries and *psb*A-*trn*H were determined from ITS and *psb*A-*trn*H sequences of the *Solidago* genus in GenBank. To further verify the species identity of our samples, we conducted a BLAST search in the database of the National Center for Biotechnology Information (NCBI) to determine the species with the most similar identities. The ribosomal ITS and chloroplast *psb*A-*trn*H of *S. decurrens*, *S. gigantea*, *S. flexicaulis*, *S. juncea*, *S. petiolaris*, *S. sempervirens*, *S. shortii*, *S. fistulosa,* and *S. simplex* were downloaded from the GenBank. The ribosomal ITS and chloroplast *psb*A-*trn*H were spliced by SequenceMatrix 1.78. In addition, we built phylogenetic trees for all 152 *S. canadensis* populations, 7 *S. altissima* populations, and closely related species with the ITS-*psb*A-*trn*H sequences, using the MEGA 4.1 Beta 3 software based on the maximum likelihood (ML) analyses, using a GTR substitution model as determined. We assessed branch support with 1000 bootstrap replicates.

### 4.4. Determination of Ploidy Level

We grew up plants from each seed family in the greenhouse. Fresh leaf material was held in self-sealing plastic bags containing silica and transported to the laboratory for ploidy level determination. We applied a modified flow cytometry method by Galbraith et al. [86] for the determination of ploidy level. Briefly, 0.5 g fresh leaf tissue was chopped up with a razor blade in 2 mL of ice-cold Galbraith buffer (PH 7.0, 45 mM MgCl_2_, 30 mM sodium citrate, 20 mM 4-morpholinepropane sulfonate, 0.5% *w*/*v* polyvinylpyrrolidone, and Triton X-100 1 mg/mL) in a vitreous Petri dish held on a chilled brick. The resultant fine slurry was filtered through a 50 μm microfilter into 2 mL polystyrene tubes and incubated at 4 °C for 5 min. The final filtrate was treated with 10 μL/mL of RNase A for 10 min at 4 °C. Then, 200 μL/mL of propidium iodide (PI) staining solution was added and incubated at 4 °C for 30 min in darkness. The fluorescence strength of PI-DNA conjugates was measured by flow cytometry (BD FACS, Becton Dickinson, Franklin Lakes, NJ, USA) using a 488 nm laser and 590/40 emission filter [87]. The fluorescence can be converted to chromosome number using standard population with known ploidy [6,76]. Therefore the native diploid population (CA09, 2n = 18) which chromosome number has been assessed using mitotic root-tip squashes [6], was used as an internal reference to measure the ploidy of ten populations. The ratio of DNA content of different ploidy was determined previously by the determination of ploidy level of a large number of populations [6]. *Glycine max* Merr. ‘Polanka is used as an internal reference to measure the DNA content of diploids [54]. Flow-cytometry data acquisition and analysis was done with the software FloMax version 2.0 (Quantum Analysis, Münster, Germany). Only histograms with CVs smaller than 5% and with a minimum peak height of 50 particles were accepted. Other histograms were discarded and corresponding samples were designated no signal [88].

### 4.5. Measurement of Morphological Traits

Populations of *S. canadensis*, *S. altissima*, *S. gigantea*, *S. simplex*, and *S. decurrens* (Appendix A) were planted in a common garden in Pailou Teaching and Research Station of Nanjing Agricultural University (32°02′ N, 118°37′ E), Nanjing, China. This region is characterized as being a subtropical monsoon climate, with an average annual rainfall of 1090 mm and the mean annual temperature is 15 °C (lowest in Feb at 2.7 °C and highest in July at 28.1 °C). They were watered, weeded, and sprayed with pesticides regularly. After plant establishment, three individuals of each population were selected, and three replicates of each individual were used to measure the 7 vegetative traits: height, stem, leaf length, leaf width, number of epidermal hairs of the main vein (per 2 mm length), number of epidermal hairs of the lateral vein (per 2 mm length), and number of epidermal hairs of the netted venation (per 4 mm^2^) (Table 3). One leaf from each plant was randomly removed from the upper, middle, and lower parts of the plant to measure the leaf length and width and epidermal hair-related traits. In the observation of leaf epidermal hairs, the apical part of the leaf, the middle part of the leaf, and the base of the leaf were measured as replicates.

During the flowering period, 56 populations of *S. canadensis* were selected, and populations of other species remain unchanged. Three individuals of each population were selected, and three replicates of each individual were used to measure the 4 floral traits: length of the ray floret, height of the involucre, number of ray florets, and number of disc florets (Table 3).

Morphological traits of populations of different sources, ploidy, and species were compared, and each morphological index was standardized by the maximum method for cluster analysis. Although a diploid population was previously found in Russia [6], it was investigated as an individual population due to its proximity to Asia and distance from other European populations.

### 4.6. Geographical Differentiation Analysis of Different Ploidy Populations

The geographical differentiation of different ploidy populations in Europe and other regions was analyzed by combining with our previous research [6] and literature records about *S. canadensis* and *S. altissima* (Appendix A; Appendix A). Climatic factors of all occurrences were extracted from worldclim (https://www.worldclim.org/ (accessed on 1 December 2021)) for the analysis of differences in climatic factors between the distribution ranges of different ploidy populations. In addition, the ecological niche differentiation pattern of different ploidy populations in Europe, North America, and Asia (the Russian population merged into Asia) are compared by comparing latitude and mean July temperatures of their distribution sites.

### 4.7. Data Analysis

The average and square deviation of morphological traits and climate factors for different geo-cytotypes of *S. canadensis* in different regions were calculated, and the significance of the differences (*p* < 0.05) between geo-cytotypes in different regions was analyzed using one-way ANOVA with Least Significant Difference (LSD). All data are expressed as the mean ± standard error (Mean ± SE). Statistical analyses and plotting were implemented in R v4.1.2 [89].

## 5. Conclusions

Our study provides morphological and molecular evidence that the hexaploidy of *S. canadensis* has successfully invaded in Europe and *S. altissima* and *S. canadensis* should be considered as one complex species. There were significant differences in morphological traits among different geo-cytotypes. Polyploidy has evolved adaptively in different introduced ranges, enhancing the competitive ability of *S. canadensis* to achieve its successful invasion. Ploidy-driven ecological niche differentiation of *S. canadensis* in the northern hemisphere depends on environmental factors, with no significant differentiation in North America and Europe, and significant differentiation in Asia, suggesting that the ploidy-driven adaptive evolution in the introduced range depends on the degree of difference from the environmental factors of the native range. Compared with the diploid, the polyploid of *S. canadensis* has a higher mean July temperature in its distribution area and is more adaptable to temperature changes. The distribution area of *S. canadensis* will gradually expand globally in the future, and the changes in the distribution area of the polyploids should be paid attention to in time.

## Figures and Tables

**Figure 1 plants-12-01113-f001:**
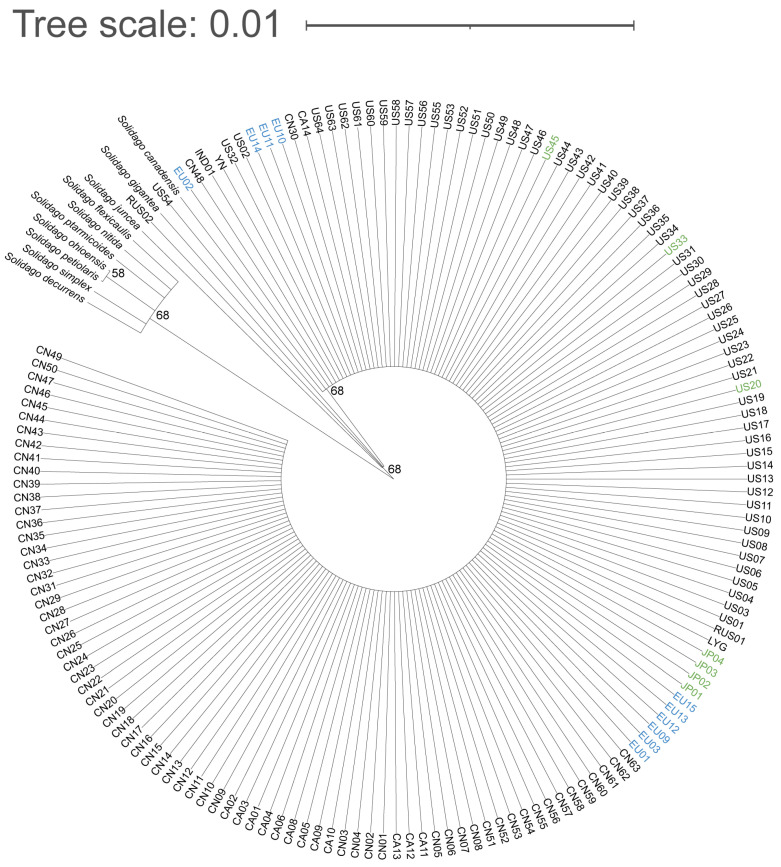
Phylogenetic tree of all *S. canadensis* populations based on ribosomal ITS and *trn*H-*psb*A intergenic spacer sequences. Blue represents Europe populations; green represents populations identified previously as *S. altissima*.

**Figure 2 plants-12-01113-f002:**
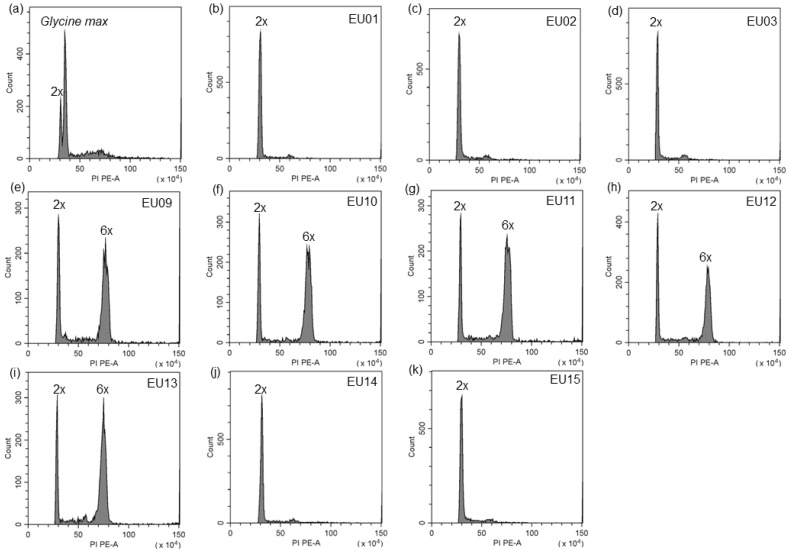
Flow cytometric ploidy analysis of populations in Europe. (**a**) *Glycine max* Merr. ‘Polanka of which DNA content is 2.5 pg [54], was used as an internal reference, therefore DNA content of the diploid population (CA09) is 2.15 pg. (**b**–**k**) The relative nuclear DNA content of each population, native diploid population (CA09) used as the internal reference.

**Figure 3 plants-12-01113-f003:**
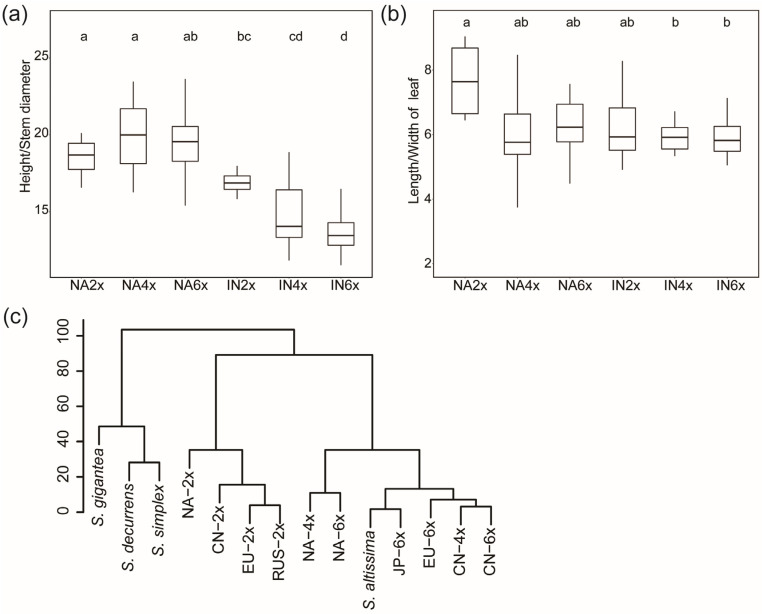
A comparison of morphological traits of different geo-cytotype populations. (**a**) The ratio of length to width, (**b**) ratio of height to stem diameter, (**c**) cluster analysis based on 11 phenotypic traits. CN represents the population of China, EU represents the population of Europe, JP represents the population of Japan, NA represents the population of North America, and RUS represents the population of Russia; *S. altissima* includes populations of North America and Japan. Different lowercase letters indicate significance.

**Figure 4 plants-12-01113-f004:**
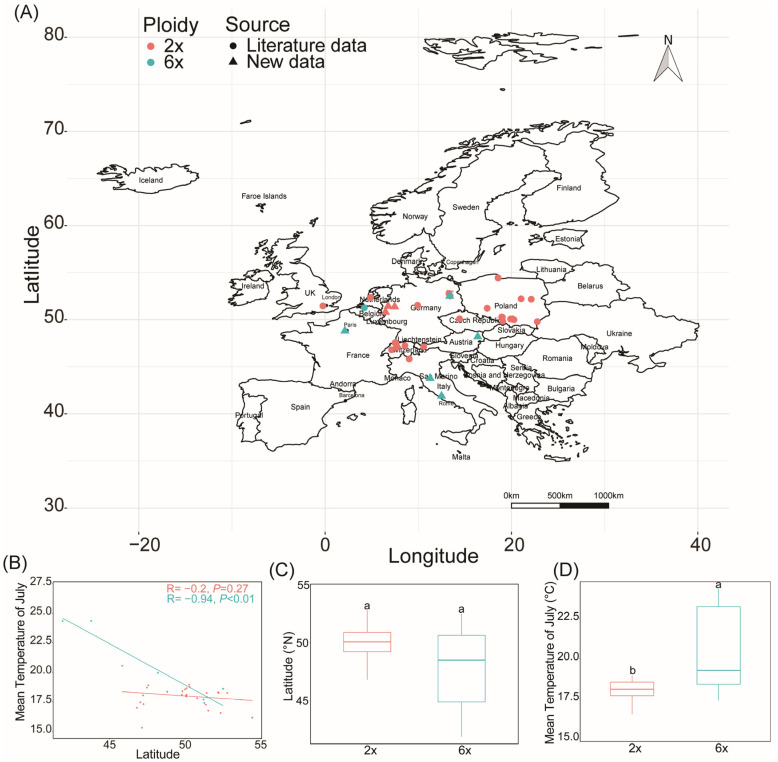
Distribution of different cytotypes of *S. canadensis* in Europe. (**A**) Distribution of different ploidy populations in Europe, (**B**) geographical differentiation of different ploidy populations, (**C**) comparison of latitudes, and (**D**) mean temperatures of July of different ploidy populations. Different lowercase letters indicate significance.

**Figure 5 plants-12-01113-f005:**
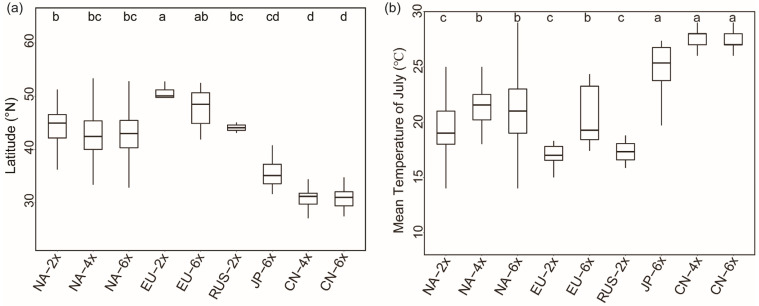
Comparison of latitude (**a**) and mean temperature of July (**b**) of different geo-cytotypes of *S. canadensis* in different regions. Note: CN represents the population of China, EU represents the population of Europe, JP represents the population of Japan, NA represents the population of North America, and RUS represents the population of Russia. Different lowercase letters indicate significance.

**Table 1 plants-12-01113-t001:** Test of ITS sequences of *Solidago canadensis* blasted in NCBI database.

Population Codes	Accession Number from Genbank	Species	Query Cover	Homology Comparison to *S. canadensis*	Variable Sites
EU01	HQ142590.1	*Solidago canadensis*	100%	100%	
EU02	HQ142590.1	*Solidago canadensis*	100%	100%	
EU03	HQ142590.1	*Solidago canadensis*	100%	100%	
EU09	MT610958.1	*Solidago canadensis*	100%	100%	
EU10	MT610958.1	*Solidago canadensis*	100%	99.85%	492 (A→G)
EU11	MT610958.1	*Solidago canadensis*	100%	99.85%	433 (T→C)
EU12	MT610958.1	*Solidago canadensis*	100%	100%	
EU13	MT610958.1	*Solidago canadensis*	100%	100%	
EU14	MT610958.1	*Solidago canadensis*	100%	99.85%	499 (C→T)
EU15	MT610958.1	*Solidago canadensis*	100%	100%	

**Table 2 plants-12-01113-t002:** Test of *psb*A-*trn*H intergenic spacer sequences of *Solidago canadensis* blasted in NCBI database.

Population Codes	Accession Number from Genbank	Species	Query Cover	Homology Comparison to *S. canadensis*	Variable Sites
EU01	KX214929.1	*Solidago canadensis*	100%	100%	
EU02	KX214929.1	*Solidago canadensis*	100%	99.53%	12 (C→-)
EU03	KX214929.1	*Solidago canadensis*	100%	100%	
EU09	KX214780.1	*Solidago canadensis*	100%	100%	
EU10	KX214780.1	*Solidago canadensis*	100%	100%	
EU11	KX214780.1	*Solidago canadensis*	100%	100%	
EU12	KX214780.1	*Solidago canadensis*	100%	100%	
EU13	KX214780.1	*Solidago canadensis*	100%	100%	
EU14	KX214780.1	*Solidago canadensis*	100%	100%	
EU15	KX214780.1	*Solidago canadensis*	100%	100%	

**Table 3 plants-12-01113-t003:** A comparison of morphological traits of different geo-cytotype populations in different regions and closely related species.

Group	Height/cm	Stem Diameter/mm	Leaf Length/cm	Leaf Width/cm	Number of Epidermal Hairs on the Main Vein per 2 mm	Number of Epidermal Hairs on the Lateral Vein per 2 mm	Number of Epidermal Hairs on the Netted Venation per 4 mm^2^	Length of the Ray Floret/mm	Height of Involucre/ mm	Number of Ray Florets	Number of Disc Florets
NA-2x	62.5 ± 2.89 c	3.39 ± 0.25 d	8.98 ± 1.61 g	1.17 ± 0.37 c	61.95 ± 10.79 a	31.22 ± 5.33 ab	53.71 ± 8.67 a	6.07 ± 0.82 bcd	3.84 ± 0.51 ef	10.5 ± 2.68 b	3.6 ± 1.28 cd
NA-4x	128.78 ± 34.86 b	6.45 ± 1.58 cd	13.12 ± 2.53 de	2.16 ± 0.46 b	58.5 ± 10.72 a	29.39 ± 4.9 b	50.9 ± 11.52 a	6.03 ± 1.27 bcd	4.72 ± 0.52 b	12.87 ± 3.3 a	4.96 ± 1.87 b
NA-6x	137.99 ± 30.06 ab	7.3 ± 1.92 cd	13.78 ± 2.67 cd	2.2 ± 0.57 ab	60.2 ± 12.03 a	32.34 ± 6.98 a	50.55 ± 12.5 a	5.9 ± 0.93 cd	3.97 ± 0.45 def	8.57 ± 1.74 cd	3.53 ± 1.34 cd
RUS-2x	62.4 ± 1.95 c	4.07 ± 0.25 cd	12.11 ± 1.4 ef	2.01 ± 0.28 bc	53.82 ± 6.76 ab	23.22 ± 3.38 c	31.48 ± 3.85 bc	5.61 ± 0.6 de	4.17 ± 0.28 cd	9.71 ± 2.25 bc	3.92 ± 1.06 c
EU-2x	64.46 ± 2.64 c	3.88 ± 0.46 d	11.21 ± 1.39 f	2.07 ± 0.42 b	55.67 ± 4.9 ab	22.29 ± 2.87 cd	33.41 ± 2.82 b	5.25 ± 0.36 e	3.57 ± 0.47 f	10.96 ± 1.65 b	3.48 ± 0.59 cd
EU-6x	141.22 ± 14.63 ab	10.72 ± 1.29 ab	15.48 ± 1.94 b	2.82 ± 0.56 a	53.75 ± 3.68 ab	20.63 ± 2.32 cd	31.31 ± 5.08 bc	5.83 ± 0.34 cde	4.13 ± 0.31 cde	10.84 ± 0.8 b	3.24 ± 0.52 cd
JP-6x	139.16 ± 5.92 ab	10.18 ± 0.55 b	14.9 ± 2.19 bcd	2.62 ± 0.72 ab	50.42 ± 12.91 b	22.61 ± 3.95 cd	19.55 ± 6.81 cd	5.38 ± 0.68 de	3.97 ± 0.61 def	9.89 ± 1.47 bc	3.45 ± 0.86 cd
CN-2x	66.54 ± 4.27 c	3.88 ± 0.6 d	7.28 ± 1.28 h	1 ± 0.15 c	42.8 ± 4.24 b	21.47 ± 4.48 cd	26.19 ± 4.01 bcd	4.82 ± 0.42 e	3.32 ± 0.31 f	9 ± 1.05 bcd	4.4 ± 0.84 bc
CN-4x	145.66 ± 21.41 a	10.05 ± 2.39 b	15.39 ± 2.23 b	2.71 ± 1.1 a	52.1 ± 15.26 b	21.55 ± 7.44 cd	26.21 ± 16.43 bcd	6.44 ± 0.65 b	4.32 ± 0.62 c	10.18 ± 1.84 bc	3.32 ± 1.1 cd
CN-6x	143.11 ± 19.5 ab	10.2 ± 2.01 b	15.88 ± 2.01 b	2.68 ± 0.47 a	50.44 ± 13.24 b	22.07 ± 6.57 cd	26.4 ± 12.18 bcd	6.08 ± 0.81 bc	4.29 ± 0.49 c	9.83 ± 1.35 bc	3.2 ± 0.87 d
*S. altissima*	137.82 ± 7.03 ab	10.45 ± 0.84 ab	14.91 ± 2.44 bc	2.62 ± 0.64 ab	50.8 ± 10.43 b	22.04 ± 4.8 cd	19.04 ± 5.89 cd	5.08 ± 0.67 de	3.74 ± 0.64 ef	9.65 ± 1.24 bc	3.59 ± 0.74 cd
*S. decurrens*	106.67 ± 4.51 cb	7.57 ± 0.5 bcd	8.71 ± 0.76 gh	2.02 ± 0.11 bc	0.73 ± 0.12 d	1.39 ± 0.65 e	1.83 ± 0.47 e	12.38 ± 0.76 a	6.86 ± 0.65 a	5.85 ± 0.31 d	13.7 ± 2.19 a
*S. gigantea*	144.11 ± 11.92 ab	12.51 ± 0.8 a	21.74 ± 2.94 a	2.87 ± 0.33 a	19.65 ± 2.22 c	14.02 ± 2.86 de	12.36 ± 1.15 d	5.31 ± 0.6 e	3.7 ± 0.36 ef	10.2 ± 1.18 bc	6.01 ± 1.02 b
*S. simplex*	112.33 ± 5.51 bc	8.57 ± 0.7 bc	7.06 ± 2.04 h	1.99 ± 0.37 bc	16.73 ± 4.1 c	10.33 ± 2.12 de	16.67 ± 7.79 cd	4.59 ± 0.3 e	4.01 ± 0.12 ef	9.18 ± 0.95 bcd	2.68 ± 0.38 d

CN represents the population of China, EU represents the population of Europe, JP represents the population of Japan, NA represents the population of North America, and RUS represents the population of Russia; *S. altissima* includes populations of North America and Japan. Different lowercase letters indicate significance.

## Data Availability

Data are available from the Dryad Digital Repository: https://datadryad.org/stash/share/i47u-h452Q7JDll8EAb4WkAtiA3WXLeAOdg3_XSl5J4 (accessed on 12 November 2022).

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
