# Peer review of "Cytogeography of Naturalized *Solidago canadensis* Populations in Europe"

_plants, 2023, doi:10.3390/plants12051113_

Round 1
Reviewer 1 Report
The manuscript Cytogeography of naturalized Solidago canadensis populations in Europe by Zhongsai Tian, Jiliang Cheng, Jingxuan Xu, Dongyan Feng, Xiaoxiao Yuan, Zheng Zhang, Yu Zhang, Zhiyuan Mao, Sheng Qiang describes some of the findings of Solidago's research.
This work raises many questions. So the measurement of ploidy and the selection of roots of another plant as a control is not correct. Removing part of the data is possible only if the complete data is available as a reference or given in the supplement, since nowhere is it indicated that this selection method corresponds to the data obtained by another method. DNA measurement does not directly correlate with ploidy. Why the chromosome count was not made is not clear.
The map is surprising, since for example there are no points on the territory of Russia, and if you have a sample from this country, all the more so different from others, it would be worth describing it.
The list of references does not contain many works related to the study of this species and its distribution in Europe, little attention is paid to the issues of hybrid forms and their differences, there are no photos of plants and botanical descriptions of samples.
Without these data, it is difficult to evaluate the work.
The main problem is either a poorly stated or erroneous method for determining ploidy and the lack of evidence of botanical differences between the samples.
I think the authors should revise the work or submit it again.
Reviewer 2 Report
Tian et al report on the cytogeography of Solidago canadensis in Europe and other continents; the manuscript is well written and the methods are soundly described, figures are mostly well done. The finding/corroboration that polyploids seem to occur in hotter climates is of general interest for a wider readership. I have only a few minor comments.
1. Since you also have studied an extensive number of populations from Asia and elsewhere, would it make change to reflect this also in the title of the paper? The title sounds like you provide a full picture of the species, but later in the text you say you would have needed more samples to get the full picture of the species in Europe.
2. The concept of "plasticity" should be mentioned and included in the discussion of the results; so far you are only speaking of the "influence of the environment".
3. There are a few language "glitches" that need to be fixed; e.g. line 344 should better read "Solidago canadensis is..."; in line 499 "the polyploid of S. canadensis" should be re-phrased.
4. In line 125 "The text continues here." should be deleted I guess.
5. In line 238 the citation "Semple(Semple and Uesugi, 2017)" should better be "Semple and Uesugi (2017) compared...".
6. Check the spacings in the citations in the text, some seem to be missing.
7. In line 460 you should refer to the SM for the table of populations otherwise it becomes unclear of what you are talking. Also check that there are references to all figures, tables and SM in the manuscript.
8. Table 1 should be formatted (broadened) to fit single lines.
Round 2
Reviewer 1 Report
The manuscript Cytogeography of naturalized Solidago canadensis populations in Europe by authors Zhongsai Tian, Jiliang Cheng, Jingxuan Xu, Dongyan Feng, Jian Zhong, Xiaoxiao Yuan, Zheng Zhang, Yu Zhang, Zhiyuan Mao , Sheng Qiang has been improved.
Changes made to the text make it possible to understand the images. Text expanded. Materials important for botanists have been added to the supliment.
Unfortunately, the strange map, which lacks images of sampling sites in the USA, Canada and Russia, has remained in the manuscript. I do not understand what the problem is in order to change it. It is now customary to indicate the digital geolocation of sampling. There might be ethical issues here.
The probabilistic assumption in the method for determining ploidy remains in question, since this method is unreliable.
Nevertheless, the authors are right to publish such works.
I can't make a definitive decision on this issue. But I think the work can be published as an option for evaluating botanical studies using cytophotometry.
